# Microlens Array Fabrication by Using a Microshaper

**DOI:** 10.3390/mi12030244

**Published:** 2021-02-28

**Authors:** Meng-Ju Lin, Cheng Hao Wen

**Affiliations:** Department of Mechanical and Computer-Aided Engineering, Feng Chia University, Taichung City 407, Taiwan; skyuonne1994331@yahoo.com.tw

**Keywords:** micro surface shaping, cutter-path planning, micro-optoelectromechanical system (MOEMS), microshaper, isoplane, nonsilicon-based micromachining

## Abstract

A simple, easy, inexpensive, and quick nonsilicon-based micromachining method was developed to manufacture a microlens array. The spherical surface of the microlens was machined using a microshaper mounted on a three-axis vertical computer numerical control (CNC) machine with cutter-path-planning. The results show the machined profiles of microlens agree well with designed profiles. The focus ability of the machined microlens array was verified. The designed and measured focal lengths have average 1.5% error. The results revealed that the focal lengths of micro lens agreed with the designed values. A moderate roughness of microlens surface is obtained by simply polishing. The roughness of the lens surface is 43 nm in feed direction (x-direction) and 56 nm in path interval direction (y-direction). It shows the simple, scalable, and reproducible method to manufacture microlenses by microshaper with cutter-path-planning is feasible.

## 1. Introduction

Micro-optoelectromechanical systems (MOEMSs) [1,2] are a critical category of microelectromechanical systems. Microlenseses are essential components used in MOEMS devices and are used in applications such as tracking [3], collimation, coupling of lights [4,5], imaging with complementary metal-oxide semiconductor and charge-coupled devices [6], imaging of confocal microscopes [7], and pop-up displays [8] because of their light focusing and collimation ability. Moreover, recent developments show microlenses have more specific and novel applications. The micro lenses can be used in Mirau interferometers [9], imaging scanners [10], integrated photonic platform [11], vertical cavity surface emitting lasers beam shaping [12], etc. Apparently, microlenses play more important role in MOEMSs.

Traditionally, techniques for manufacturing microlenses often involved silicon-based micromachining. Microlenses can be fabricated by injecting droplets from microjets and solidifying the droplets to form lens surfaces [13]. Photomask lithography with LIGA-like manufacturing is also used to fabricate microlenses [14]. Furthermore, reactive-ion etching with modified parameters [15] and the reflow method are used to fabricate microlenses. The photoresist is heated to its liquid phase and allowed to flow into a curved surface under capillary force [16]. By using a cross-linked network of photoactive polymers, microlens fabrication was performed with an optically induced dielectrophoresis system [17].

In these methods, reflow methods have well developed. The microlenses of OCT system made by photoresist reflow and ICP plasma etching processes have good quality of small sizes and roughness [18]. Reflow of glass on silicon cavity can form convex lens and assembled with the other planar lens by anodic bonding. It would generate lens doublets with good quality [19].

Reflow and selective etching can fabricate microlenses used in THz antenna integrated heterodyne array [20]. Silicon collimating microlenses made by reflow and RIE have high numerical aperture for mid-infrared quantum cascade lasers [21]. Combining the dose-modulated lithography and reflow process, high quality aspheric microlens array are successfully fabricated [22]. Combining reflow, ultraviolet nanoimprint lithography, and replica mold processes, micro-lens array with antireflection structure can be fabricated [23]. The silicon solid immersion lenses for mid-infrared imaging can be made by nanoimprint used PDMS stamp and reflow [24]. PDMS nanoimprint and reflow can also fabricate well-defined shape microlenses [25].

Silicon-based micromachining has the advantages of batch fabrication for mass production, small dimensions, and integration with electronic devices. However, as designing devices and producing prototype, it needs cyclically modified. And in each modification, only several devices are made. These several devices of prototype often do not need massive production. And in each round of silicon-based micromachining, it may spend several days due to preparing photo mask for lithography and some processes needing vacuum environment. Therefore, if a simple, easy, and quick fabricate method could be used to fabricate microlenses. It would help the design and prototype modification.

In conventional mechanical machining, complex three-dimensional curved surfaces can be effectively manufactured due to well develop of CNC machine. Milling is often used for machining three-dimensional curved surfaces. Efficient machining can be achieved through ball-end milling on multiaxis CNC machines [26,27,28,29,30,31]. Complex or free form surfaces can be precisely machined through profile milling of work pieces on a CNC machine with path-planning [32,33,34,35,36,37,38]. To improve the machining of complex surfaces through CNC milling, parameter tuning and path-planning were investigated [39]. Furthermore, the effect of milling machining on roughness was explored [40]. Although milling with path-planning has the advantages of complex-surface fabrication, the constraints of the knife radius limit path-planning applications for micro components. Therefore, the development of specific methods is necessary for fabricating surface profiles with small dimensions. Moreover, mechanical machining is hard compatible to integrated circuit (IC). Therefore, mechanical machining micro components need specific techniques to be compatible with IC.

In this study, a simple and easy method was proposed for fabricating microlens arrays. By this developed nonsilicon-based micromachining method, producing the prototype of micro lens does not need to prepare masks used in lithography. The method also has no vacuum processes like depositions to produce devices. It could be fast, easily, and inexpensively fabricated. Moreover, due to path planning, specific curves and surfaces of MEMS devices can be made. It would help the optical devices design. Instead of a milling knife, a microshaper with a small nose radius (as shown in Figure 1 and Figure 2) was mounted on a CNC machine. It has the advantages for easily manufacturing micro lenses with specific profiles. Combined with path-planning, the curved surfaces of a microlens were successfully fabricated using the microsphere. The machined profiles agreed with the designs. The microlens could effectively focus light, and the measured focal length was in agreement with the theoretical focal length calculated from the design. By simply polishing, the results also show the roughness can reduce to about fifty nano meters. It is feasible to manufacture micro lenses easily and simply by method developed in this work. And it also shows that this manufacture method has potential to be used in manufacturing aspheric microlens array or micro convex-concave lenses. It can further more be used in microscraping of MEMS devices. And this method could be used to manufacture stamps of nanoimprint. Therefore, it can further more be used in micromachining over silicon wafer.

## 2. Materials and Methods

In this work, spherical microlenses were machined due to spherical surfaces often used in convex lenses. As displayed in Figure 3, the function of spherical surface is expressed as follows:(1)$$Z=-R+\sqrt{R^{2}-q^{2}}$$
where q $=\;\sqrt{x^{2}+y^{2}}$ is the distance measured from the symmetric axis of the convolute surface, *Z* is the profile height measured from the q axis, and R is the radius of curvature of the apex of the aspheric surface.

In optical properties of microlenses, the focal length is important and considered in this work. The focal length of convex lens was designed using the following equation [41]:(2)$$\frac{1}{f}=\left(\frac{n}{n_{0}}-1\right)\left[\frac{1}{R_{1}}-\frac{1}{R_{2}}\right]$$
where *n* and *n_0_* are the indexes of refraction of the lens and air, respectively, and *R_1_* and *R_2_* are the radii of curvature of the first and second surfaces, respectively. For a planoconvex lens, the radius of curvature *R_2_* of the second surface is infinite. Therefore, Equation (2) can be expressed as follows:(3)$$\frac{1}{f}=\left(\frac{n}{n_{0}}-1\right)\frac{1}{R_{1}}$$

For air and PMMA, the indexes of refraction are 1 and 1.49, respectively.

For measuring focal length effectively, a simple method is used two conjugates. As displayed in Figure 4, the focal length *f* can be determined through two conjugates of the lens. When lens is at the two conjugates, the well-known two conjugate method states that the object would have clear images and the distance between object and image is fixed. In Figure 4, when the lens is at either of the two conjugates, which are separated by a distance *d*, the distance between the object and the image *l* would be a fixed value. Therefore, the focal length can be calculated using the following equation:(4)$$f=\frac{l^{2}-d^{2}}{4l}$$

Traditionally, CNC machine with cutter path planning was used to machine complex surface shape because of its excellent surface cutting ability. Therefore, it is used to manufacture spherical microlens in this study. However, because the microlens is on a micrometer scale, a special knife is required. As displayed in Figure 1, a stiff tungsten steel microshaper with a small cutting nose radius was mounted on a CNC machine for use as the machining knife. Furthermore, this knife could also be coated with diamond layer on the tip. Therefore, some brittle and hard materials could be scraped [42]. For example, wedge-type light guide plate of efficiency of a LED backlight module. The knife has length of 50 mm and diameter of 4 mm. The knife nose radius was 50 μm (Figure 2). Due to remove materials by the knife being not similar to traditional machining milling, the cutter path planning of CNC machine would be different. And its machining properties of roughness is also different. Conventionally, for ball-end milling, the machining paths are orthogonal to the machining directions when the materials are removed or cut. Unlike milling, the machining direction of microshaper cutting materials is the same as the machining path. Because the nose tip of the microshaper (Figure 2) scrapes the material, the nose tip of the microshaper should be in the forward direction in the tool path. Because machining differs considerably between ball-end milling and microshaper scraping, the cutter-path planning of the microshaper is different in ball-end milling. For the novel microshaper, the path-planning was nonparametric. For nonparametric path-planning, isoplane, isolevel, isoscallop, and isoparametric methods have been developed [37]. Because of the specific cutting characteristic of the microshaper mounted on a three-axes CNC machine, the isoplane method is simple and was therefore preferred for path-planning in this study. As displayed in Figure 5, the feed direction of the knife was the x-direction. As the knife moved forward in the x-direction, the microshaper changed the scraping height (z-direction) to obtain the desired surface profile (Figure 5). After finishing this path, the microshaper returned to its initial position without machining any materials and moved a path interval in the y-direction to the next path. Then the microshaper will scrape the materials on next path.

The motion of the microshaper in the x-z plane is displayed in Figure 5b. The microshaper performed x- and z-axis movements simultaneous to obtain the curved surface. However, when the microshaper moved an interval in the y-direction to the machine in the next path, as displayed in Figure 5a, scallops were produced between two paths. Figure 6 displays the scallops between every adjacent paths. The scallop is an error induced by the geometry of knife nose radius and path interval. It would induce larger roughness. The relation of scallop height, path interval, and radius of knife nose can be expressed as the following equation [37]:(5)$$\Delta y=\sqrt{8r\varepsilon \;\;}$$
where ∆y is the path interval, *r* is the nose radius of the microshaper, and *ε* is the scallop height. The path interval ∆y is defined in programming the CNC machine cutter path planning. Therefore, the scallop height can be estimated: $\varepsilon =\frac{{\left(\Delta y\right)}^{2}}{8r}$. Therefore, there would be larger roughness in path interval direction (*y*-direction).

In this study, a planoconvex lens array was designed and manufactured. The profile height Z (Figure 3) was used to determine the height of the microshaper in the cutting path (Figure 4b). To fabricate the microlens array, a three-axis vertical CNC machine (CPV-750, Campro Precision Machinery Co., Ltd., Taichung, Taiwan) with a resolution of 1 μm was used. A 1-mm-thick polymethyl methacrylate (PMMA) substrate was used to manufacture the microlens on this substrate due to its good optical properties. The CNC programming elaborates on Appendix A.

## 3. Results and Discussions

The microlens array was successfully fabricated, as displayed in Figure 7. Figure 7a reveals the array of 5 × 6 micro spherical lenses of 4-mm radius of curvature. It costs about an hour and half to manufacture the array of 5 × 6 micro spherical lenses by CNC machine with cutter-path planning. Figure 7b,c illustrates the cross section of a 1-mm and 4-mm radius of curvature microspherical lens respectively. The radii of curvature *R* of Figure 7b,c calculated from Equation (1) are 1.07 and 4.04 mm. They agreed the design values well. Figure 8 illustrates the image of the cross section observed on microscope with scale grid on eyepiece. We add an arc of a circle to compare the profile of the machined convex surface. It shows the profiles of the micro surface is enveloped by the arcs. Figure 9a,b show the profiles of machined surface compares with the theoretical profiles calculated from Equation (1). From the results, it shows that the profiles of cross sections machined by micro shaper agree well with the design profiles.

To observe whether the micro lens can focus light or not, the lens is put in front of a He-Ne laser as shown in Figure 10. The lens focus of the laser in Figure 10. As shown in Figure 7b,c, it reveals that the microlens successfully focused light. This verified the focusing ability of the lens.

Machining properties would also affect the optical properties of microlenses. To evaluate the properties of material scraping by the microshaper, a surface roughness (SJ-310, Mitutoyo, Kanagawa, Japan) was used to measure the roughness of lens surface. The roughness measured is arithmetic mean deviation roughness. The roughness in feed direction (x-direction) is 116 nm. And the roughness in path interval direction (y-direction) is 462 nm. After machined by using CNC, the roughness of the microlenses is large. Therefore, the microlenses needs furthermore polished. To reduce the roughness, a simple and not expensive method is used. Spreading the toothpaste on soft cloth and rubbing the PMMA substrate, the microlenses can be polished by the abrasive containing in toothpaste. After polishing, using water to wash out toothpaste, the roughness reduces to moderate values of 43 nm and 56 nm in feed direction and path interval direction respectively. Comparing with specific method of thermal radiation induced local reflow (TRILR) [43] for reducing roughness of PMMA, this roughness value is acceptable.

Figure 11 displays setup of the focal length measurement, where *A* is a screen for observing the image, *B* is the microlens, and *C* is a light with a luminous flux of 1000 lm as being the object expressed in Figure 4. Initially, the distance between the screen A and light C is fixed. Then move the microlens to find the two conjugate positions for observing clear image projecting on screen as displaying in Figure 12. The distance of the two conjugates is measured used vernier caliper.

The measured focal length was compared with the designed values as shown in Table 1 and Figure 13. Table 1 compares the focal lengths of the design and the measured results. In Table 1, *f_m_* is the average value of focal length measured from the 5 × 6 micro spherical lenses. And the standard errors of the 30 lenses for different R are also illustrated. The largest error between the designed and measured focal length is 2.9%. The average error is 1.5%. These errors may be induced from machining processes and measurement. Due to reducing machining time, the cutter offset often considered in traditional milling processes is ignored and there is no cutter path compensation in path planning. It would also induce error between design and measurement focal length. And from the standard errors, the measured uncertainty is acceptable. As the cross section of microleses shown in Figure 7, the radii of curvature (*R*) calculated from Equation (1) are 1.07 and 4.04 mm respectively. Apply these values to Equation (3), the focal lengths calculated are 2.16 and 8.24 mm respectively. They are larger than *f_d_* and *f_m_*. As shown in Figure 13, for different radii of curvature (R), the measured focal lengths agree well with design focal lengths. The coefficients of the determination (r-squared, *R*^2^) is 0.9988. It shows that the microlenses made by microscraping with path planning having good focal length precision.

## 4. Conclusions

A simple, easy, and inexpensive method to fabricate microlens array through nonsilicon-based micromachining by using microshaper on CNC machine with path-planning is developed and proved to be feasible. The microlens array was successfully fabricated. The profile of cross section of the microlens shows a round arc and agrees well with design curve. For optical properties, the average error between designed and measured focal lengths is 1.5%. The coefficients of the determination (r-squared) for designed and measured focal lengths is 0.9988 for micro-lens arrays with 13 different radii of curvature. The measured focal lengths of the microlenses were in agreement with the designed focal lengths. The results revealed that the microlens array exhibited good focus ability. After simple polishing, the roughness is 43 nm and 56 nm in feed direction and path interval direction respectively. It shows the machining method developed in this work has good machining properties. The microlens array fabrication method using microshaper mounted on CNC machine with path-planning has good machining and optical properties. Therefore, the micro-lens manufacturing method developed in this work reveals nice results and has advantages for quick, simple, easy, and low cost prototypes manufacturing. Moreover, the results show the profiles of lens are precisely machined used this method. The method can be furthermore used in manufacturing aspheric microlens array or micro convex-concave lenses and stamps of nanoimprint.

## Figures and Tables

**Figure 1 micromachines-12-00244-f001:**
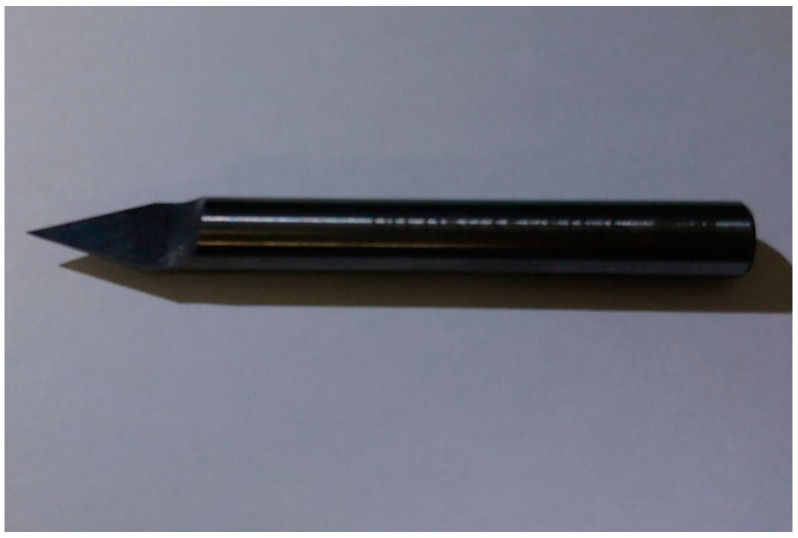
Tungsten steel microshaper. The microshaper in mounted on CNC machine to machine the microlens.

**Figure 2 micromachines-12-00244-f002:**
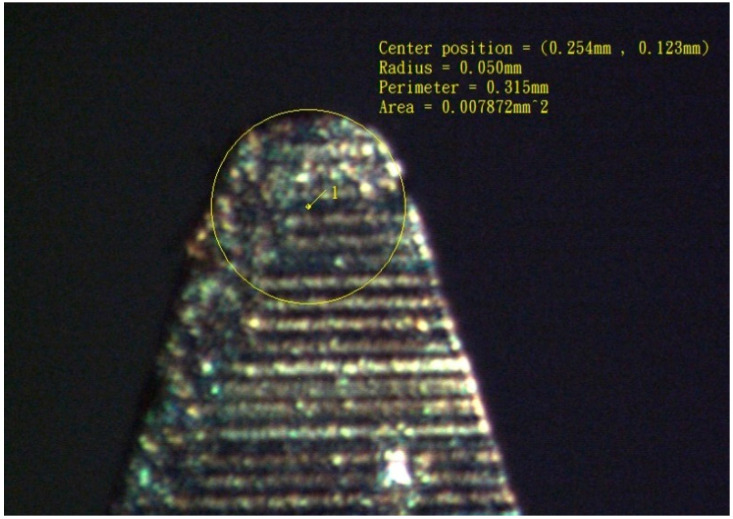
Nose radius of the microshaper measured on a microscope. The knife radius has radius of 50 μm.

**Figure 3 micromachines-12-00244-f003:**
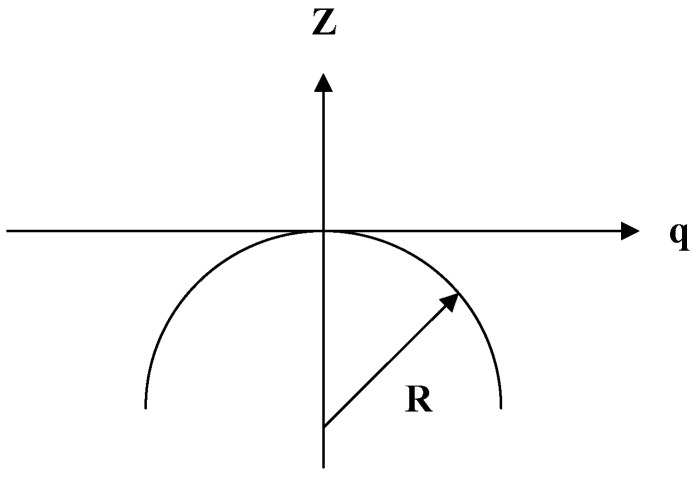
Spherical curve dimensions.

**Figure 4 micromachines-12-00244-f004:**
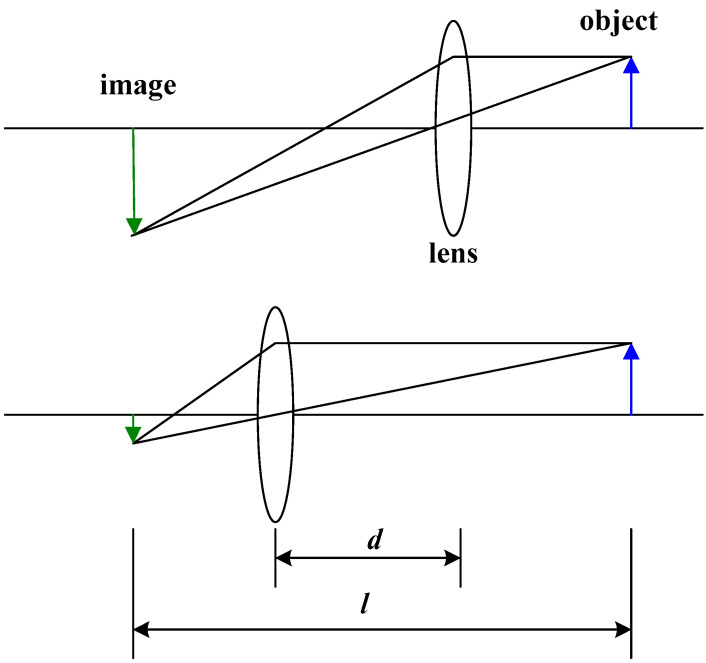
Focal length measurement using two conjugates method.

**Figure 5 micromachines-12-00244-f005:**
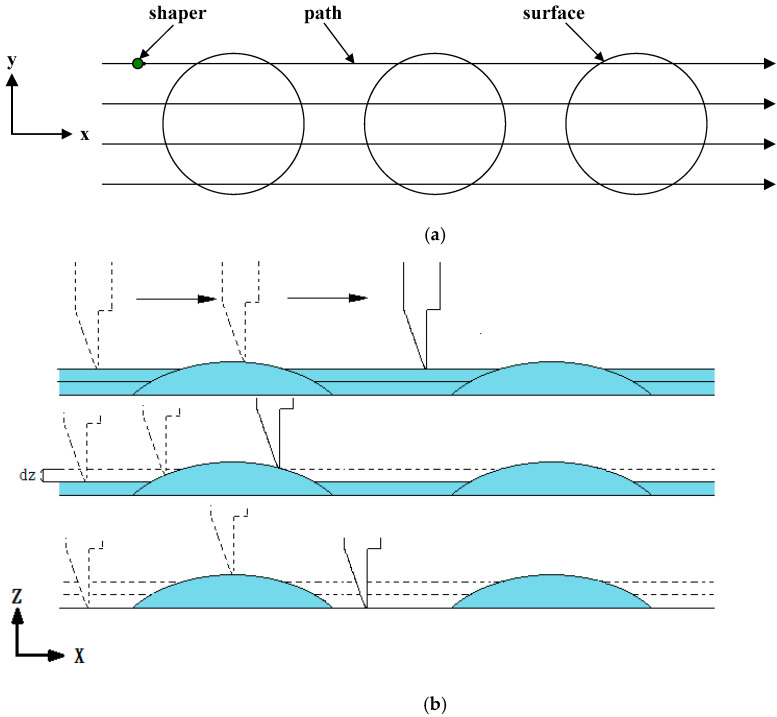
Cutting path of the microshaper: (**a**) x-y plane and (**b**) x-z plane. The x-direction is the feed direction of the microshaper. As the shaper finishing machining the path in feed direction, it will move to next path in y direction and continue to machine.

**Figure 6 micromachines-12-00244-f006:**
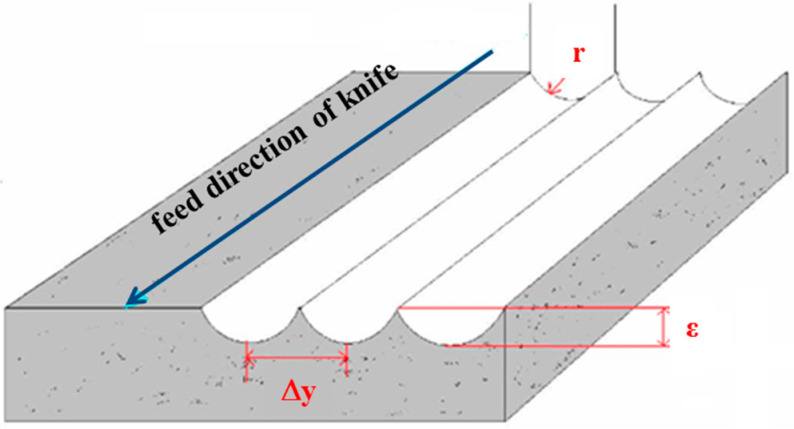
Scallop will happen between machined path intervals.

**Figure 7 micromachines-12-00244-f007:**
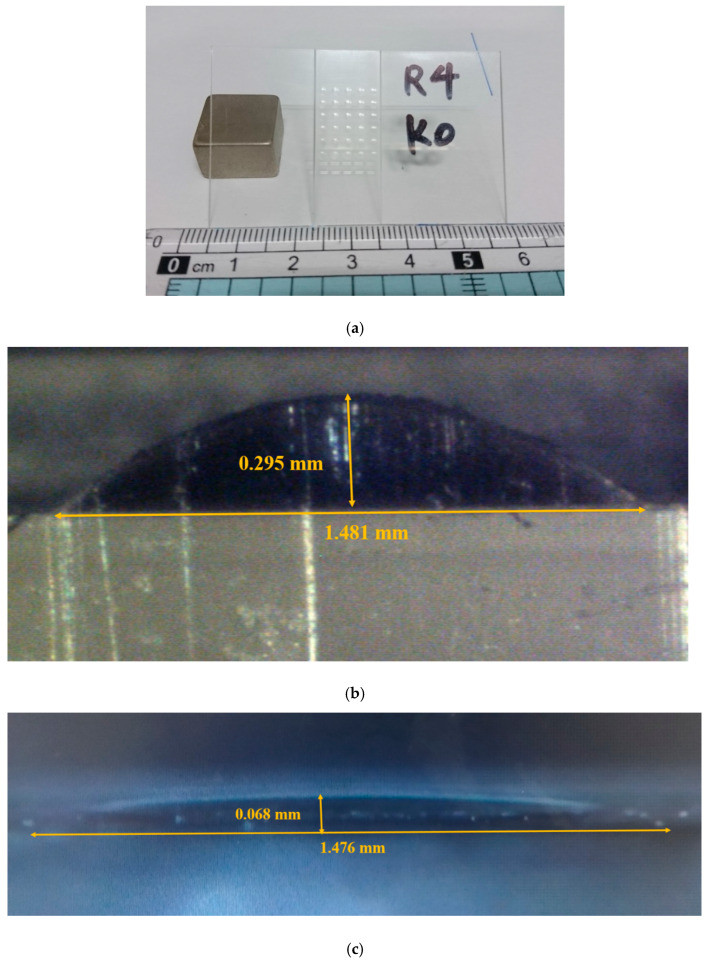
Microlens array fabrication results, (**a**) the array of 5 × 6 micro spherical lenses, (**b**) cross section of the fabricated microlens with *R* = 1 mm and (**c**) cross section of the fabricated microlens with *R* = 4 mm.

**Figure 8 micromachines-12-00244-f008:**
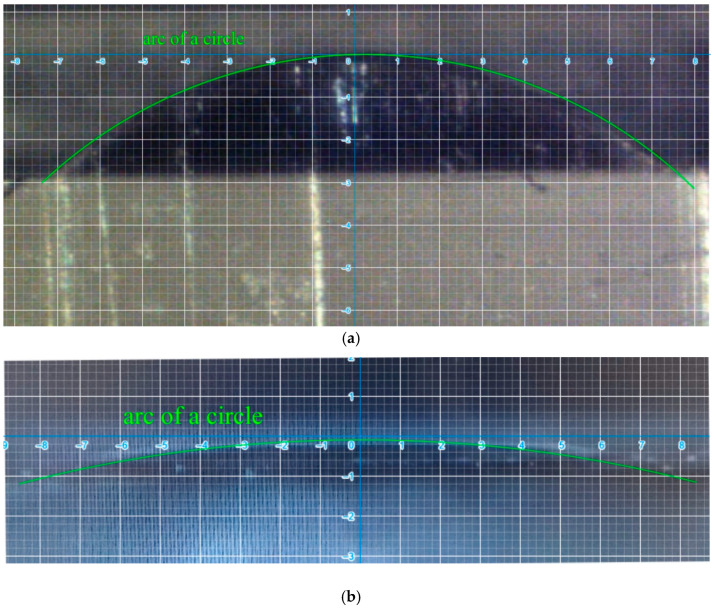
Cross section observed on microscope with scale grid on eyepiece. (**a**) *R* = 1 mm. (**b**) *R* = 4 mm.

**Figure 9 micromachines-12-00244-f009:**
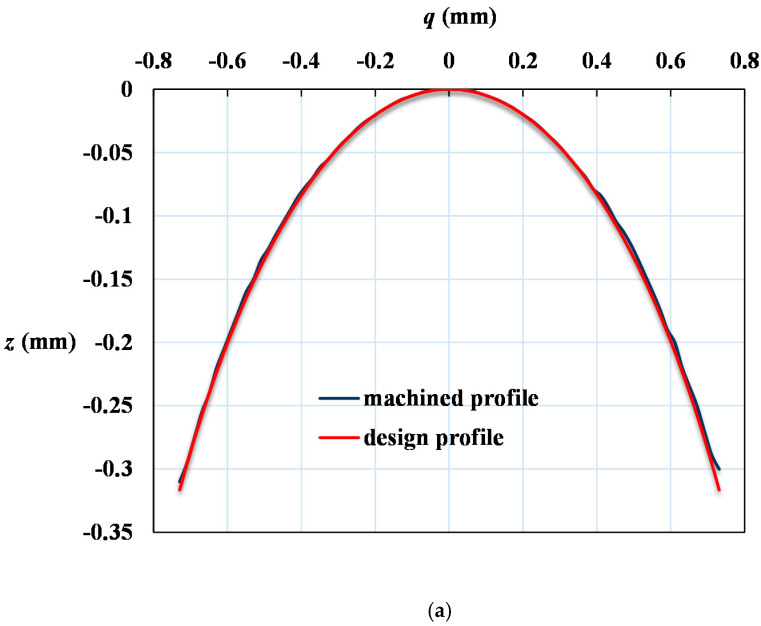
Comparison of machined curved surface profiles with theoretical conic section profiles calculated from Equation. (**a**) *R* = 1 mm. (**b**) *R* = 4 mm.

**Figure 10 micromachines-12-00244-f010:**
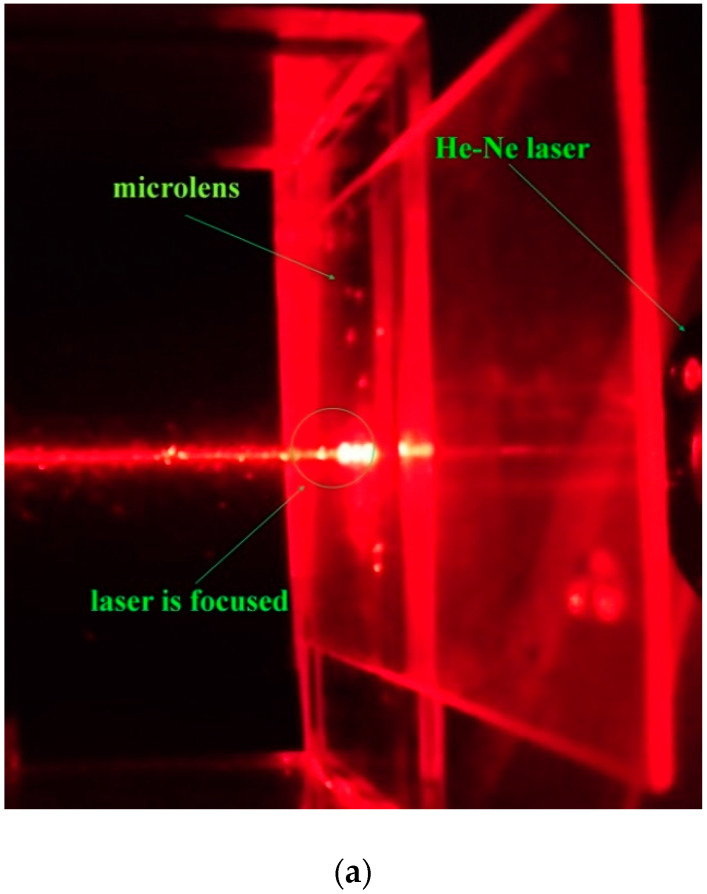
(**a**) The He-Ne laser is focused by using the microlens. (**b**) The light spot of laser beam without passing through lens projects on screen. (**c**) The light spot of laser beam passing through lens projects on screen.

**Figure 11 micromachines-12-00244-f011:**
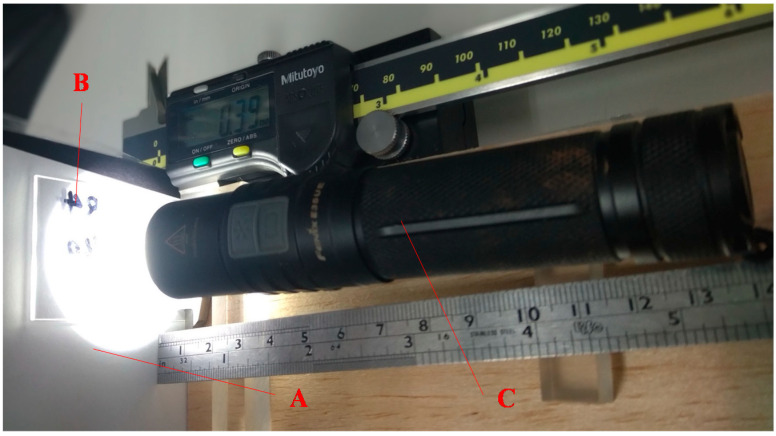
Focal length measurement of the microlens, where A is the screen, B is the microlenses, and C is the light respectively.

**Figure 12 micromachines-12-00244-f012:**
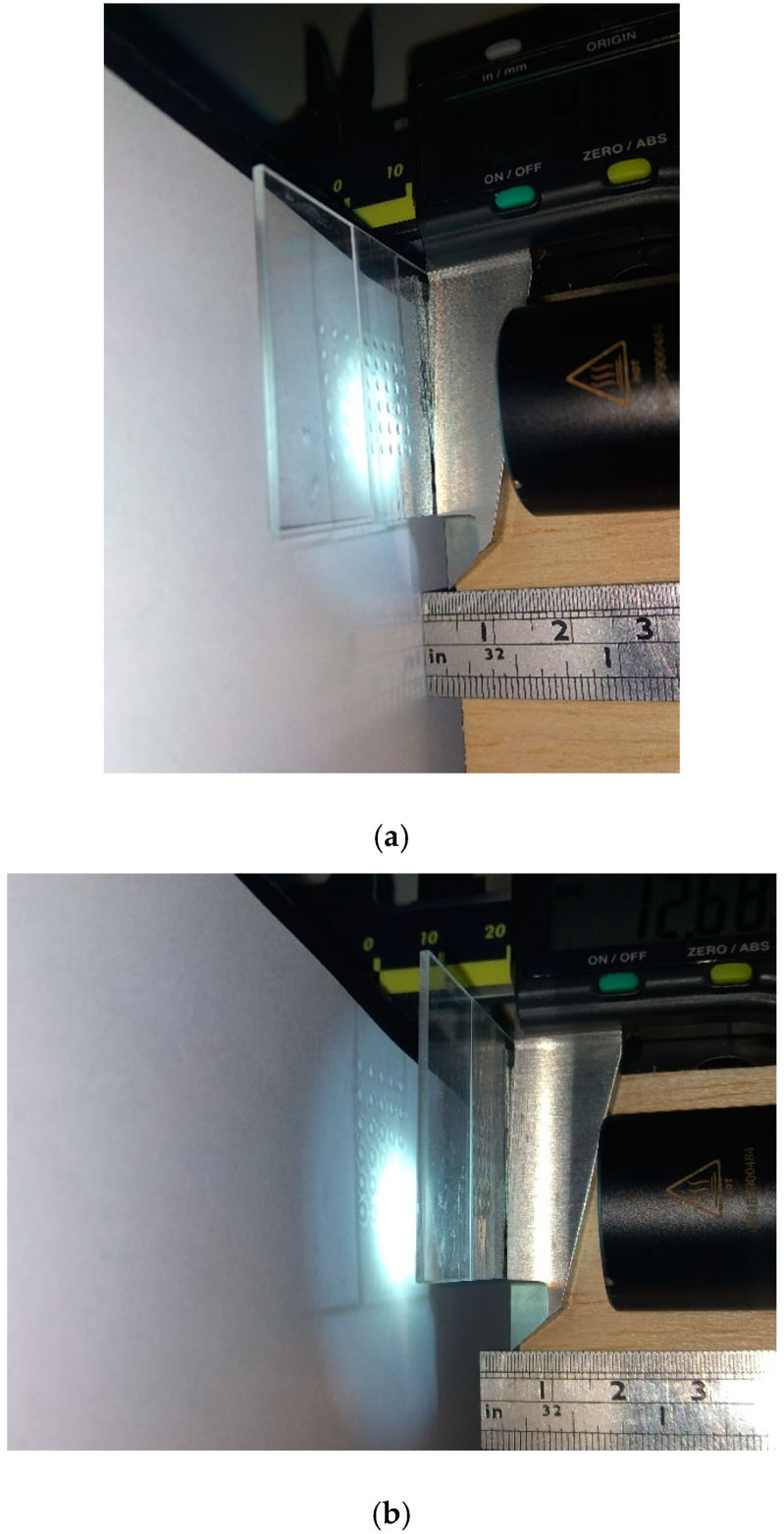
Observe image projecting on the screen to find the two conjugates. (**a**) The position of first conjugate. (**b**) The position of the second conjugate.

**Figure 13 micromachines-12-00244-f013:**
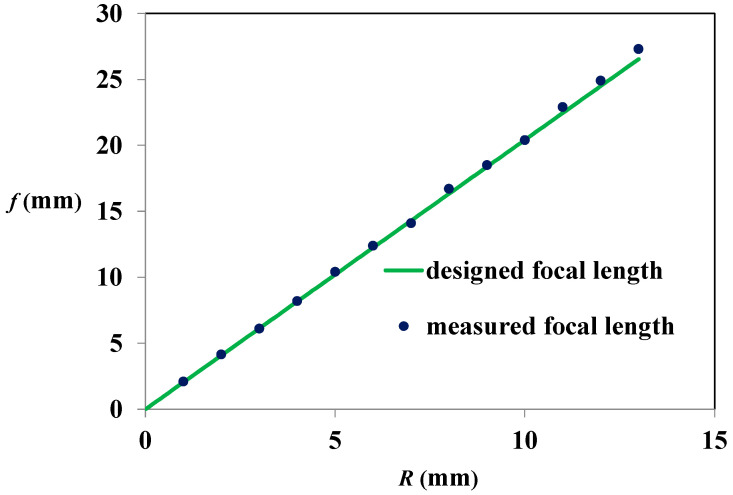
Relation of focal length (*f*) and radius of curvature (*R*).

**Table 1 micromachines-12-00244-t001:** Comparison of the measured and designed focal lengths, where *R* is radius of curvature, *f_d_* is the designed focal length, and *f_m_* is the measured focal length respectively.

*R* (*mm*)	*f_d_* (*mm*)	*f_m_*(*mm*)	Standard Error of *f_m_* (mm)	Error (%)
1	2.04	2.10	0.04	2.9
2	4.08	4.14	0.04	1.5
3	6.12	6.22	0.04	1.6
4	8.16	8.20	0.09	0.5
5	10.20	10.41	0.08	2.0
6	12.24	12.40	0.06	1.3
7	14.28	14.15	0.08	−0.9
8	16.32	16.67	0.08	2.1
9	18.36	18.49	0.10	0.7
10	20.40	20.37	0.06	−0.1
11	22.44	22.84	0.10	1.8
12	24.48	24.88	0.06	1.6
13	26.53	27.13	0.10	2.3

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
