# Peer review of "Microlens Array Fabrication by Using a Microshaper"

_micromachines, 2021, doi:10.3390/mi12030244_

Round 1

Reviewer 1 Report

Dear authors,

This is a significant improvement compared to the previous version of this manuscript. The results are much better presented and explained. However, in my opinion, extensive English editing is needed in order to improve the style and make the manuscript easy to follow. Please consult someone who is English proficient to help you with the text. Especially, the introduction needs big improvement, not in terms of content (content is good) but in terms of style.

Here are some of my comments:

Comment 1:

Line 27: You wrote: "imaging of confocal microscopes" It should be "imaging with confocal microscopes" or simply " confocal microscopy".

Line 29: Should be: "Mirau interferometers" instead of "Mirau interference devices"

Line 30: "Imaging scanners" instead of "scanner for imaging solution"

Comment 2:

Lines 56-57 are in contradiction to line 53.

Comment 3:

Please organize your paragraphs correctly!!! It is a general remark but in the intro obvious that your paragraphs are too long and a reader is getting lost in all the information. My suggestion would be:

  • State of the art technics for microlens fabrication and group them
    • Reflow
    • Reflow+RIE
    • Stamping 
    • etc
  • Advantages and disadvantages of each group
  • CNS summary
  • Summary of your paper (what the reader will see in Section 2, section 3 and etc)

Each point should be at least 1 paragraph...

Comment 4:

Equation 1 for me is not really clear... First, I would choose the centre of the sphere as a coordinate 0.0 position. In that case, Z = sqrt ( R2 - q2) If you want to put the top of the sphere as 0.0 than Z =  sqrt ( R2 - q2) - R.  Please explain how did you come up with your formula.

Comment 5:

Do you have an image without focusing the laser light to give as a comparison to image 10? It would be nice to see the difference between focused an unfocused beam. You can simply shoot the light next to your lens, just through the material.

Comment 6:

Could you elaborate on your polishing method? You say the toothpaste is used to smoothen the surface. Does the toothpaste have the same refractive index as PMMA? How does that affect the lens quality if two materials are of different n?

Comment 7:

Could you present photos of both conjugates in Fig 12?

Comment 8:

Conclusion of the scientific results is not only a summary of what is written. An author must have some answers to the research questions introduced in the manuscript. Please, provide a proper conclusion to this manuscript, which has really nice results. It would be a pity to have such a weak conclusion (i.e. a simple summary of what is done).

Please comment on your methodology and quality of results presented. What this can mean for future developments in this field? What does it mean to have 43 and 56 nm of surface roughness. Any many more comments can be made based on the presented results.

Comment 9

Also, I did not see anything about how you programmed CNC. Could you elaborate on that as well?

Author Response

This is a significant improvement compared to the previous version of this manuscript. The results are much better presented and explained. However, in my opinion, extensive English editing is needed in order to improve the style and make the manuscript easy to follow. Please consult someone who is English proficient to help you with the text. Especially, the introduction needs big improvement, not in terms of content (content is good) but in terms of style.

Here are some of my comments:

Comment 1:

Line 27: You wrote: "imaging of confocal microscopes" It should be "imaging with confocal microscopes" or simply " confocal microscopy".

Line 29: Should be: "Mirau interferometers" instead of "Mirau interference devices"

Line 30: "Imaging scanners" instead of "scanner for imaging solution"

Our reply

Thanks a lot for reviewer’s comments. We have changed the expressions.

Comment 2:

Lines 56-57 are in contradiction to line 53.

Our reply

Thanks a lot for reviewer’s comments. We are sorry for expressing not accurate statement. It is rewritten as shown in lines 57-59.

Comment 3:

Please organize your paragraphs correctly!!! It is a general remark but in the intro obvious that your paragraphs are too long and a reader is getting lost in all the information. My suggestion would be:

  • State of the art technics for microlens fabrication and group them
    • Reflow
    • Reflow+RIE
    • Stamping 
    • etc
  • Advantages and disadvantages of each group
  • CNS summary
  • Summary of your paper (what the reader will see in Section 2, section 3 and etc)

Each point should be at least 1 paragraph...

Our reply

Thanks a lot for reviewer’s comments. We have re-modified the paragraphs as reviewer’s suggestion.

Comment 4:

Equation 1 for me is not really clear... First, I would choose the centre of the sphere as a coordinate 0.0 position. In that case, Z = sqrt ( R2 - q2) If you want to put the top of the sphere as 0.0 than Z =  sqrt ( R2 - q2) - R.  Please explain how did you come up with your formula.

Our reply

Thanks a lot for reviewer’s comments. This equation is modified from implicit function of conical curve for cutter-path planning in CNC machining:

where, k is the conic constant and k = 0 for sphere. For avoiding misleading, we express the equation as:

Comment 5:

Do you have an image without focusing the laser light to give as a comparison to image 10? It would be nice to see the difference between focused an unfocused beam. You can simply shoot the light next to your lens, just through the material.

Our reply

Thanks a lot for reviewer’s comments. We add two photos to compare the light spot of laser beam in front and behind the microlens as shown in Figure 10 (b) and (c).

Comment 6:

Could you elaborate on your polishing method? You say the toothpaste is used to smoothen the surface. Does the toothpaste have the same refractive index as PMMA? How does that affect the lens quality if two materials are of different n?

Our reply

Thanks a lot for reviewer’s comments. The PMMA is polished by abrasive containing in the toothpaste. After polishing, the toothpaste is washed. We have rewritten the expressions as shown in lines 241-244.

Comment 7:

Could you present photos of both conjugates in Fig 12?

Our reply

Thanks a lot for reviewer’s comments. We have presented both the conjugates as shown in the Figure 12 (a) and (b).

Comment 8:

Conclusion of the scientific results is not only a summary of what is written. An author must have some answers to the research questions introduced in the manuscript. Please, provide a proper conclusion to this manuscript, which has really nice results. It would be a pity to have such a weak conclusion (i.e. a simple summary of what is done).

Please comment on your methodology and quality of results presented. What this can mean for future developments in this field? What does it mean to have 43 and 56 nm of surface roughness. Any many more comments can be made based on the presented results.

Our reply

Thanks a lot for reviewer’s comments. We have re-modified the conclusion as your suggestion.

Comment 9

Also, I did not see anything about how you programmed CNC. Could you elaborate on that as well?

Our reply

Thanks a lot for reviewer’s comments. We elaborate the CNC programming on Appendix.

Reviewer 2 Report

The authors developed a simple method to fabricate microlen arrays. It showed that the CNC machined curvature was in agreement with that predicted by theoretical equation. Overall, the discussion is comprehensive and results gathered seem informative. However, the following comments should be addressed before this manuscript could be considered for publication.

  1. In INTRODUCTION section, the authors mentioned that silicon-based micromachining is time-consuming and proposed the present approach is “quick”. However, no evidence was given to prove its efficiency of manufacturing. For example, how many hours/minutes did it take to fabricate an array of 5x6 micro spherical lenses like Figure 7a?

  1. Page 2, lines 61 and 63. The abbreviation of CNC should be defined at first mention (line 61).

  1. The demonstration of focusing ability shown in Figure 10 is a good idea but the image is vague. I suggest to provide a more specific evidence (for example, the comparison of beam sizes in front of and behind the microlen).

  1. English should be improved. For example, what did “literal results” mean?

Author Response

The authors developed a simple method to fabricate microlen arrays. It showed that the CNC machined curvature was in agreement with that predicted by theoretical equation. Overall, the discussion is comprehensive and results gathered seem informative. However, the following comments should be addressed before this manuscript could be considered for publication.

In INTRODUCTION section, the authors mentioned that silicon-based micromachining is time-consuming and proposed the present approach is “quick”. However, no evidence was given to prove its efficiency of manufacturing. For example, how many hours/minutes did it take to fabricate an array of 5x6 micro spherical lenses like Figure 7a?

 Our reply

Thanks a lot for reviewer’s comments. It costs an hour and half to manufacture the array. It can be found in lines 199-200. We are sorry for expressing not accurate statement about “quick”. It means the CNC machining does not need some time-consuming processes in clean room. For example, the masks used in lithography need several days to produce. And some processes like deposition is time-consuming to reach the vacuum condition. We rewrite the expression as shown in lines 57-59.

Page 2, lines 61 and 63. The abbreviation of CNC should be defined at first mention (line 61).

 Our reply

Thanks a lot for reviewer’s comments. We have moved the abbreviation to its first mention.

The demonstration of focusing ability shown in Figure 10 is a good idea but the image is vague. I suggest to provide a more specific evidence (for example, the comparison of beam sizes in front of and behind the microlen).

Our reply

Thanks a lot for reviewer’s comments. We add two photos to compare the light spot of laser beam in front and behind the microlens as shown in Figure 10 (b) and (c).

English should be improved. For example, what did “literal results” mean?

Our reply

Thanks a lot for reviewer’s comments. We are sorry for expressing not accurate statement and try to improve the English. We have rewritten the expression as shown in lines 245-246.

This manuscript is a resubmission of an earlier submission. The following is a list of the peer review reports and author responses from that submission.

Round 1

Reviewer 1 Report

Dear authors,

The presented lens fabrication method looks very simple, scalable, and reproducible. Indeed, the MOEMS filed can benefit from more simplified lens fabrication techniques, thus making this manuscript important for the field. Therefore, the presentation and the motivation for this work must be improved.

Please rewrite the manuscript in such a way that you follow rules for paragraphs. All your sections are maxed 2 paragraphs. It is very hard to read and follow the information. In the end, give someone who is not related to this topic to read the manuscript and see if the person can understand it nicely.

Comment 1:

Some images are too big and occupy a lot of space thus pushing the text to the next page. This sometimes makes it hard to flow the text. For example, Fig 1 and 2 could be placed next to each other. Then, Fig 6a and b could be next to each other. Moreover, you can crop only the are of interest, the reader will not care too much about the black area which represents nothing. Focus only on the lens. The comment applies to all images.

Comment 2:

Please improve the captions of images. Some explanations should also be in the caption, not only in the text, so the reader has proper info together with the image.

Comment 3:

The statement in lines 33-35 is not completely true. Indeed one can argue the expense of microfabrication processes and the cost of process development, but once developed, the results are very reproducible.

Your reference list is outdated and must be refreshed with more recent work for microlenses, in general. Please include the following related work and try to expand the list:

a) N. Llombart et al., "Silicon Micromachined Lens Antenna for THz Integrated Heterodyne Arrays," in IEEE Transactions on Terahertz Science and Technology, vol. 3, no. 5, pp. 515-523, Sept. 2013, doi: 10.1109/TTHZ.2013.2270300.

b) Eric Logean, et al., "High numerical aperture silicon collimating lens for mid-infrared quantum cascade lasers manufactured using wafer-level techniques," Proc. SPIE 8550, Optical Systems Design 2012, 85500Q (18 December 2012);https://doi.org/10.1117/12.981165

c) A. Jovic, et al., "Fabrication process of Si microlenses for OCT systems," Proc. SPIE 9888, Micro-Optics 2016, 98880C (27 April 2016);https://doi.org/10.1117/12.2227898

d) Shengzhou Huang, et al., Fabrication of high quality aspheric microlens array by dose-modulated lithography and surface thermal reflow, Optics & Laser Technology, Volume 100, 2018, Pages 298-303, ISSN 0030-3992.

e) Qiu, Jinfeng, et al. "Fabrication of microlens array with well-defined shape by spatially constrained thermal reflow." Journal of Micromechanics and Microengineering 28.8 (2018): 085015.

f) Lee, G.J., et al., "Design and Fabrication of Microscale, Thin-Film Silicon Solid Immersion Lenses for Mid-Infrared Application". Micromachines 2020, 11, 250.

g) Nicolas Passilly, et al., "Wafer-level fabrication of arrays of glass lens doublets," Proc. SPIE 9888, Micro-Optics 2016, 98880T (27 April 2016);https://doi.org/10.1117/12.2228833

In these references, scientists used different techniques and sometimes their combinations. Keep in mind that bigger lenses can easily be made using reflow on thicker polymers. One can check also the final surface quality of fabricated lenses which is a very important optical parameter.

Comment 4:

The motivation for MOEMS lenses should be updated with recent relevant work as well (lines 20-23). Some of the presented references in Comment 3 also include direct application, while some lenses were later applied to new devices.

a) Sylwester Bargiel, et al., "Vertical comb-drive microscanner with 4x4 array of micromirrors for phase-shifting Mirau microinterferometry," Proc. SPIE 9890, Optical Micro- and Nanometrology VI, 98900D (26 April 2016);

b) A. Jovic et al., "A highly miniturized single-chip MOEMS scanner for all-in-one imaging solution," 2018 IEEE Micro Electro Mechanical Systems (MEMS), Belfast, 2018, pp. 25-28, doi: 10.1109/MEMSYS.2018.8346472.

c) A. Jovic, et. al., "Self-aligned micro-optic integrated photonic platform," Appl. Opt. 59, 180-189 (2020)

d) V. Bardinal, et. al, "Advances in Polymer-Based Optical MEMS Fabrication for VCSEL Beam Shaping," in IEEE Journal of Selected Topics in Quantum Electronics, vol. 21, no. 4, pp. 41-48, July-Aug. 2015, Art no. 2700308, doi: 10.1109/JSTQE.2014.2369743.

The list can be further expanded if one follows citations of a paper. 

Comment 5:

Motivate the application of your microlenses. What was the end goal? Assembly? Microscraping over the wafer? Please elaborate and summarize it in the manuscript.

Comment 6:

If the goal (Comment 5) is to implement only a cheaper method, please describe how would you apply such a lens in one MOEMS device? What would be the advantages and disadvantages of such lens?

Comment 7:

Why are you presenting R=4 mm lenses in Fig 6 a and R=1 mm in Fig 6b. Where is R= 1mm in Table 1?

Comment 8:

Lines 103-105 described the tool for fabrication (CNC machine) should be in materials and methods.

Comment 9:

The measurement setup description is poorly described. Please describe nicely and in full detail the measurement method and setup.

Comment 10:

How can one understand from image 7 that the lens is focusing? Please describe your results in images properly. Feel free to use labels in Fig 7 as well as you did in Fig 8. It can be more than the just letters A, B, C. Can be: light source, light spot etc. 

Comment 11: 

Improve result presentation of table 1 by adding the error margin between designed and measured in %. The table could have the format:

R [mm] | fd [mm] | fm [mm] | Error [%]

Comment 12:

Combine 3. Result and 4. Discussion into 3. Results in Discussion. There is no need to have it separetly. 

Comment 13:

What is the final surface roughness of your lens? Is it 6 um? Please measure it and elaborate on the results. Describe how will it influence the lens quality.

Comment 14:

Honestly, very poor conclusion. There is no comparison of your lenses with other lenses made in the field. There is no comparison of fabrication costs (why not?) and etc. The conclusion is just a brief mentioning of what is done, no any scientific conclusion. This point is a very big minus for the manuscript.

Author Response

Reviewer 1

Dear authors,

The presented lens fabrication method looks very simple, scalable, and reproducible. Indeed, the MOEMS filed can benefit from more simplified lens fabrication techniques, thus making this manuscript important for the field. Therefore, the presentation and the motivation for this work must be improved.

Please rewrite the manuscript in such a way that you follow rules for paragraphs. All your sections are maxed 2 paragraphs. It is very hard to read and follow the information. In the end, give someone who is not related to this topic to read the manuscript and see if the person can understand it nicely.

Our reply

Thanks a lot for reviewer’s comments. We are sorry for our presentation of this manuscript. We have tried our best to re-modify this work and express more readable and acceptable.

Comment 1:

Some images are too big and occupy a lot of space thus pushing the text to the next page. This sometimes makes it hard to flow the text. For example, Fig 1 and 2 could be placed next to each other. Then, Fig 6a and b could be next to each other. Moreover, you can crop only the are of interest, the reader will not care too much about the black area which represents nothing. Focus only on the lens. The comment applies to all images.

Our reply

Thanks for reviewer’s comments. We have cut out the dark parts of the figures and images.

Comment 2:

Please improve the captions of images. Some explanations should also be in the caption, not only in the text, so the reader has proper info together with the image.

Our reply

Thanks for reviewer’s comment. We have written more explanations to make the caption be understood easily.  

Comment 3:

The statement in lines 33-35 is not completely true. Indeed one can argue the expense of microfabrication processes and the cost of process development, but once developed, the results are very reproducible.

Your reference list is outdated and must be refreshed with more recent work for microlenses, in general. Please include the following related work and try to expand the list:

  1. a) N. Llombart et al., "Silicon Micromachined Lens Antenna for THz Integrated Heterodyne Arrays," in IEEE Transactions on Terahertz Science and Technology, vol. 3, no. 5, pp. 515-523, Sept. 2013, doi: 10.1109/TTHZ.2013.2270300.

  1. b) Eric Logean, et al., "High numerical aperture silicon collimating lens for mid-infrared quantum cascade lasers manufactured using wafer-level techniques," Proc. SPIE 8550, Optical Systems Design 2012, 85500Q (18 December 2012);https://doi.org/10.1117/12.981165

  1. c) A. Jovic, et al., "Fabrication process of Si microlenses for OCT systems," Proc. SPIE 9888, Micro-Optics 2016, 98880C (27 April 2016);https://doi.org/10.1117/12.2227898

  1. d) Shengzhou Huang, et al., Fabrication of high quality aspheric microlens array by dose-modulated lithography and surface thermal reflow, Optics & Laser Technology, Volume 100, 2018, Pages 298-303, ISSN 0030-3992.

  1. e) Qiu, Jinfeng, et al. "Fabrication of microlens array with well-defined shape by spatially constrained thermal reflow." Journal of Micromechanics and Microengineering 28.8 (2018): 085015.

  1. f) Lee, G.J., et al., "Design and Fabrication of Microscale, Thin-Film Silicon Solid Immersion Lenses for Mid-Infrared Application". Micromachines 2020, 11, 250.

  1. g) Nicolas Passilly, et al., "Wafer-level fabrication of arrays of glass lens doublets," Proc. SPIE 9888, Micro-Optics 2016, 98880T (27 April 2016);https://doi.org/10.1117/12.2228833

In these references, scientists used different techniques and sometimes their combinations. Keep in mind that bigger lenses can easily be made using reflow on thicker polymers. One can check also the final surface quality of fabricated lenses which is a very important optical parameter.

Our reply

Thanks for reviewer’s comments. We have referred these papers. And in our manuscript, we add the sentences ” In these methods, reflow methods have well developed. The microlenses of OCT system made by photoresist reflow and ICP plasma etching processes have good quality of small sizes and roughness [18]. Reflow of glass on silicon cavity can form convex lens and assembled with the other planar lens by anodic bonding. It would generate lens doublets with good quality [19]. Reflow and selective etching can fabricate microlenses used in THz antenna integrated heterodyne array [20]. Silicon collimating microlenses made by reflow and RIE have high numerical aperture for mid-infrared quantum cascade lasers [21]. Combining the dose-modulated lithography and reflow process, high quality aspheric microlens array are successfully fabricated [22]. Combining reflow, ultraviolet nanoimprint lithography, and replica mold processes, micro-lens array with antireflection structure can be fabricated [23].  The silicon solid immersion lenses for mid-infrared imaging can be made by nanoimprint used PDMS stamp and reflow [24]. PDMS nanoimprint and reflow can also fabricate well-defined shape microlenses [25].” to express their importance. It can be found in lines 39-50.

We are sorry for expressing not accurate statement of lines 33-35. Such expression would make misunderstanding. We are apology for bad writing. We want describe even silicon-based micromachining can produce low cost devices for massive production. But it is really a budget burden for a laboratory in university like us. We delete these sentences and write “However, as designing devices and producing prototype, it needs cyclically modified. And in each modification, only several devices are made. It is not suitable for massive production of silicon-based micromachining. And in each round of silicon-based micromachining, it may spend several days. Therefore, if a simple, easy, and quick fabricate method could be used to fabricate microlenses. It would help the design and prototype modification.” I think this explain is better. It can be found in lines 52-57.

Comment 4:

The motivation for MOEMS lenses should be updated with recent relevant work as well (lines 20-23). Some of the presented references in Comment 3 also include direct application, while some lenses were later applied to new devices.

  1. a) Sylwester Bargiel, et al., "Vertical comb-drive microscanner with 4x4 array of micromirrors for phase-shifting Mirau microinterferometry," Proc. SPIE 9890, Optical Micro- and Nanometrology VI, 98900D (26 April 2016);

  1. b) A. Jovic et al., "A highly miniturized single-chip MOEMS scanner for all-in-one imaging solution," 2018 IEEE Micro Electro Mechanical Systems (MEMS), Belfast, 2018, pp. 25-28, doi: 10.1109/MEMSYS.2018.8346472.

  1. c) A. Jovic, et. al., "Self-aligned micro-optic integrated photonic platform," Appl. Opt. 59, 180-189 (2020)

  1. d) V. Bardinal, et. al, "Advances in Polymer-Based Optical MEMS Fabrication for VCSEL Beam Shaping," in IEEE Journal of Selected Topics in Quantum Electronics, vol. 21, no. 4, pp. 41-48, July-Aug. 2015, Art no. 2700308, doi: 10.1109/JSTQE.2014.2369743.

The list can be further expanded if one follows citations of a paper.

Our reply:

Thanks for reviewer’s comments. We have referred these papers and add the sentences “Moreover, recent developments show microlenses have more specific and novel applications. The micro lenses can be used in Mirau interference devices [9], scanner for imagine solution [10], integrated photonic platform [11], vertical cavity surface emitting lasers beam shaping [12], etc. Apparently, microlenses play more important role in MOEMSs.” to express. It can be found in lines 28-31.

Comment 5:

Motivate the application of your microlenses. What was the end goal? Assembly? Microscraping over the wafer? Please elaborate and summarize it in the manuscript.

Our reply

Thanks for reviewer’s comment. Our work is developed a simple and easy microlens manufacture method. It can help in help the design and prototype modification of microlenses. It can be found in lines 10 and 31. For the applications, we express as the sentences ” And it also shows that this manufacture method has potential to be used in manufacturing aspheric microlens array or micro convex-concave lenses. Maybe it can further more be used in microscraping over silicon wafer.”  It can be found in lines 77-80.

Comment 6:

If the goal (Comment 5) is to implement only a cheaper method, please describe how would you apply such a lens in one MOEMS device? What would be the advantages and disadvantages of such lens?

Our reply:

Thanks for reviewer’s comment. We explain the advantages as “It has the advantages for easily manufacturing micro lenses with specific profiles.” It can be found in lines 71-72. For disadvantages, we explain as “Moreover, mechanical machining is hard compatible to integrated circuit (IC). Therefore, mechanical machining micro components need specific techniques to be compatible with IC.” It can be found in lines 67-69.

Comment 7:

Why are you presenting R=4 mm lenses in Fig 6 a and R=1 mm in Fig 6b. Where is R= 1mm in Table 1?

Our reply

Thanks for reviewer’s comment. Fig. 6 (b) shows the lens with radius of 1 mm due to our thinking that it is clearer. We change the cross section image of lens to R = 4mm.

Comment 8:

Lines 103-105 described the tool for fabrication (CNC machine) should be in materials and methods.

Our reply

Thanks for reviewer’s comment. We move the sentences describe CNC machine and PMMA substrate to materials and methods. It can be found in lines 145-148.

Comment 9:

The measurement setup description is poorly described. Please describe nicely and in full detail the measurement method and setup.

Our reply

Thanks for reviewer’s comment. We have rewritten the measurement method and setup by adding sentences “When lens is at the two conjugates, the well-known two conjugate method states that the object would have clear images and the distance between object and image is fixed.” in lines 180-181 and “Figure 9 displays setup of the focal length measurement, where A is a screen for observing the image, B is the microlens, and C is a light with a luminous flux of 1000 lm as being the object expressed in Figure 8. Initially, the distance between the screen A and light C is fixed. Then move the microlens to find the two conjugate positions for observing clear image projecting on screen as displaying in Fig. 10. The distance of the two conjugates is measured used vernier caliper.” In lines 186-190.

Comment 10:

How can one understand from image 7 that the lens is focusing? Please describe your results in images properly. Feel free to use labels in Fig 7 as well as you did in Fig 8. It can be more than the just letters A, B, C. Can be: light source, light spot etc.

Our reply

Thanks for reviewer’s comment. We have added some words in the image to express the focus of the laser.

Comment 11:

Improve result presentation of table 1 by adding the error margin between designed and measured in %. The table could have the format:

R [mm] | fd [mm] | fm [mm] | Error [%]

Our reply

Thanks for reviewer’s comment. We have changed the format of the table.

Comment 12:

Combine 3. Result and 4. Discussion into 3. Results in Discussion. There is no need to have it separately.

Our reply

Thanks for reviewer’s comment. We have combined “3. Results” and “4. Discussions” to become “3. Results and discussions”.

Comment 13:

What is the final surface roughness of your lens? Is it 6 um? Please measure it and elaborate on the results. Describe how will it influence the lens quality.

Our reply

Thanks for reviewer’s comment. We have added the roughness measurement. It is not 6 mm. The 6 mm is the path interval of the cutter path planning when the scallop height is set for 0.1 mm. We are sorry for not express clearly and making misleading. Therefore, we rewrite the sentences as ‘’The scallop is an error induced by the geometry of knife nose radius and path interval. The relation of scallop height, path interval, and radius of knife nose can be expressed as the following equation [29]:

                                                                                                           (1)

where Dy is the path interval, r is the nose radius of the microshaper, and ε is the scallop height. The path interval Dy is defined in programming the CNC machine cutter path planning. Therefore, the scallop height can be estimated: .” It can be found in lines 125-131. The roughness measurement is expressed “Machining properties would also affect the optical properties of microlenses. To evaluate the properties of material scraping by the microshaper, a surface roughness (TR-200D, Beijing TIME High Technology Ltd, Beijing, China) was used to measure the roughness of lens surface. The roughness measured is arithmetic mean deviation roughness. The roughness in feed direction (x-direction) is 105 nm. And the roughness in path interval direction (y-direction) is 454 nm.” It can be found in lines 172-176.

Comment 14:

Honestly, very poor conclusion. There is no comparison of your lenses with other lenses made in the field. There is no comparison of fabrication costs (why not?) and etc. The conclusion is just a brief mentioning of what is done, no any scientific conclusion. This point is a very big minus for the manuscript.

Our reply

Thanks for reviewer’s comment. Honestly, because we do not really manufacture micro lenses by silicon-based micromachining, we cannot compare the cost with our micro scraping cost. Considering the cost of another devices of my previous work made by semiconductor fabrication processes (one mask photolithography, SOI wafer, RIE for 20 mm, and wet etching releasing) being about 50000 NT (new Taiwan dollars), the cost of micro lenses is less than 100 NT. That is why I say this method is cheap. However, it is not suitable to be comparing in the manuscript. Therefore, we rewrite the conclusion “A microlens array was successfully fabricated through nonsilicon-based micromachining. The results revealed that the microlens array exhibited good focus ability. The average error between designed and measured focal lengths is 2.1 %. The measured focal lengths of the microlenses were in agreement with the designed focal lengths. The roughness is 105 nm and 454 nm in feed direction and path interval direction respectively. It is feasible to manufacture microlenses simply and easily by using microshaper on CNC machine with path-planning .” It can be found in lines 217-222.

Reviewer 2 Report

From a manufacturing engineering standpoint the authors describe an interesting approach for the production of microlenses. It is conceivable that the method might have interesting applications. Also, the method as described appears to be appropriate; although some parts of the description are rather trivial.

However, the documentation of the obtained results in the presented manuscript is entirely inacceptable! Here are some out of many points:

- no evaluation of the manufactured geometry is presented whatsoever. Figure 6b is by far not suffivient for this purpose. For an optical component an analysis of both, the precise geometry and the obtained surface properties would be essential.

- no reliable documentation of the optical properties is shown. This applies for example to lens errors and the imaging quality.

- Equation (3) is obviously incorrect (f on both sides of the equation) and cannot be found in reference [29] as stated in the manuscript.

- The optical setup in Figure 8 and Figure 9 is obviously unsuitable for a precise measurement of the focal length and does not match at all with the descriptions in reference [29]

In conclusion, the presented manuscipt is not suitable for publication and should be rejected.

Author Response

Reviewer 2

From a manufacturing engineering standpoint the authors describe an interesting approach for the production of microlenses. It is conceivable that the method might have interesting applications. Also, the method as described appears to be appropriate; although some parts of the description are rather trivial.

However, the documentation of the obtained results in the presented manuscript is entirely inacceptable! Here are some out of many points:

- no evaluation of the manufactured geometry is presented whatsoever. Figure 6b is by far not sufficient for this purpose. For an optical component an analysis of both, the precise geometry and the obtained surface properties would be essential.

Our reply

Thanks for reviewer’s comment. We have added an arc of circle to envelop the cross section as shown in Fig. 6 (b). And we add the sentences “Machining properties would also affect the optical properties of microlenses. To evaluate the properties of material scraping by the microshaper, a surface roughness (TR-200D, Beijing TIME High Technology Ltd, Beijing, China) was used to measure the roughness of lens surface. The roughness measured is arithmetic mean deviation roughness. The roughness in feed direction (x-direction) is 105 nm. And the roughness in path interval direction (y-direction) is 454 nm.” To express roughness.  It can be found in lines 172-176.

- no reliable documentation of the optical properties is shown. This applies for example to lens errors and the imaging quality.

Our reply

Thanks for reviewer’s comment. We have expressed the lens error in Table 1.

- Equation (3) is obviously incorrect (f on both sides of the equation) and cannot be found in reference [29] as stated in the manuscript.

- The optical setup in Figure 8 and Figure 9 is obviously unsuitable for a precise measurement of the focal length and does not match at all with the descriptions in reference [29]

Our reply

Thanks for reviewer’s comment. We are sorry for typo of the equation. And we have corrected this equation. And the tow conjugates method is a well-known focal length measurement method. It is not a precise measurement method as the reference [29]. It is also expressed as simple method in the reference [29]. We refer the paper [29] due to its mentioning the Equation. We write a sentence “For measuring focal length effectively, a simple method is used two conjugates.” It can be found in lines 178-179. For avoiding mislead, we remove this reference.

In conclusion, the presented manuscipt is not suitable for publication and should be rejected.

Our reply

We are very sorry for worse presenting. And we have tried our best to re-modify this work and express more readable and acceptable. Please give us opportunity to publish this paper.

Round 2

Reviewer 1 Report

Dear authors,

There is a significant improvement compared to the first version but still, the text is very hard to follow. Please use paragraphs properly! Find another person who will help you to improve the style! It is already required from you in the first review version.

Honestly, there should be a lot of work done for this manuscript. Here are some extra comments which can help, but my suggestion is to start all over again and make the proper structure and add additional characterization.

Motivation is still weak (although intro is much improved) and there is no clear description of how it could be potentially used. Also, the conclusion is very weak.

Comment 1:

It is very hard to follow the text in chapter 2, the first paragraph. Please correct the whole paragraph and split it into several (see general comment).

Comment 2:

Center all equations and make some space between eq and the text. Please, format the text nicely.

Comment 3:

Figure 6 does not correspond to the lens of R = 4 mm of the radius. If one uses the measured values given in Fig 6, one gets R = 1.587 mm. How do you explain that?

Comment 4:

Please provide calculations of your R and f based on the cross-section values for your lenses. Then, compare it with f theoretical and f measured.

Comment 5:

The error of your measurements: Could you please explain the error of your measurement results, i.e. measurement uncertainty. The presented error is just a difference between measured and calculated values. However, you never expressed your measurement uncertainty.

Comment 5:

Elaborate on how does surface roughness influences lens quality. For visible, 105 and 435 are pretty bad values.

Comment 6:

You introduced the resolution of the CNC and explained the epsilon (from EQ3) but I cannot correlate it to any optical quality. Elaborate on that.

Reviewer 2 Report

Dear authors,

although you made numerous changes in your manuscript, your methods of documenting the lens geometry and surface as well as the optical properties obtained by your process remain highly inadequate. Therefore, in my opinion, the manuscript should be rejected.